# FLASH: LATENT-AWARE SEMI-AUTOREGRESSIVE SPECULATIVE DECODING FOR MULTIMODAL TASKS

## ABSTRACT

Large language and multimodal models (LLMs and LMMs) exhibit strong inference capabilities but are often limited by slow decoding speeds. This challenge is especially acute in LMMs, where visual inputs typically comprise more tokens with lower information density than text — an issue exacerbated by recent trends toward finer-grained visual tokenizations to boost performance. Speculative decoding has been effective in accelerating LLM inference by using a smaller draft model to generate candidate tokens, which are then selectively verified by the target model, improving speed without sacrificing output quality. While this strategy has been extended to LMMs, existing methods largely overlook the unique properties of visual inputs and depend solely on text-based draft models. In this work, we propose **FLASH** (Fast Latent-Aware Semi-Autoregressive Heuristics), a speculative decoding framework designed specifically for LMMs, which leverages two key properties of multimodal data to design the draft model. First, to address redundancy in visual tokens, we propose a lightweight latent-aware token compression mechanism. Second, recognizing that visual objects often co-occur within a scene, we employ a semi-autoregressive decoding strategy to generate multiple tokens per forward pass. Experiments show that FLASH consistently outperforms prior speculative decoding approaches in both unimodal and multimodal settings, achieving up to **2.68×** speed-up on video captioning and **2.55×** on visual instruction tuning tasks compared to the original LMM.

## 1 INTRODUCTION

Large multimodal models (LMMs) (Achiam et al., 2023; Team et al., 2024; Yang et al., 2024; Liu et al., 2023) have made significant progress on tasks like video captioning and visual question answering by leveraging synergistic relationships between data types. Recent studies show that processing more input tokens during inference improves contextual understanding (Muennighoff et al., 2025; Liu et al., 2025). Consequently, LMMs increasingly adopt finer-grained patch division strategies, which significantly expand the token count. This issue is particularly severe for video inputs, as the large number of the video frames leads to a significant expansion of the visual input scale. While larger models and longer contexts improve accuracy and flexibility, they also raise deployment challenges due to hardware constraints and increased computational costs.

To address the efficiency issue, recent works propose token compression to remove some visual tokens (Zhang et al., 2025; Wen et al., 2025). Since the visual tokens contain superfluous or nonessential information, pruning these tokens makes shorter input contexts and thus accelerating the generation. However, this simplification comes with potential drawbacks. For example, it can be difficult to accurately identify which tokens to prune, as seemingly non-critical tokens may actually encode latent task-specific cues (Wen et al., 2025). Therefore, while token compression offers efficiency gains, it may damage the performance by oversimplifying the visual input, especially in fine-grained multimodal tasks.

Speculative decoding offers a promising solution for accelerating inference without sacrificing performance. As an effective strategy for speeding up decoding in Large Language Models (LLMs) (Bachmann et al., 2025; Li et al., 2024b; Cai et al., 2024; Fu et al., 2024), it employs a lightweight draft model to quickly generate candidate token sequences, which are then verified in parallel by the target model through **a single forward pass**. Crucially, by applying specific acceptance-rejection criteria,

the accepted tokens can be regarded as samples from the target model's distribution, thereby preserving output quality. This fail-safe property ensures that speculative decoding maintains the accuracy of the target model while significantly reducing inference latency. However, the overall speed-up is closely linked to the acceptance rate of the candidates generated by the draft model. This relationship establishes a critical trade-off in the design of draft models: overly simplistic architectures, while computationally efficient, risk producing low-quality candidates that are frequently rejected during verification, diminishing overall latency gains.

Given that LMMs frequently leverage LLMs as their decoders, prior work (Gagrani et al., 2024) has attempted to directly transplant speculative decoding techniques from LLMs to the multimodal settings. However, this method trains a separate draft model using *only textual inputs*, which introduces additional computational overhead and ignores the visual modality. To overcome this drawback, early efforts Ganesan et al. (2025); Hu et al. (2025) have explored incorporating both visual and textual information into the draft model. However, in contrast to LLMs, LMMs contain numerous redundant visual tokens, making a naive extension of this framework to multimodal input computationally expensive. To address this limitation, we propose FLASH (Fast Latent-Aware Semi-Autoregressive Heuristics), a novel method for efficient multimodal speculative decoding that achieves a favorable trade-off between inference speed and draft quality.

In this work, we leverage two distinctive properties of multimodal data to enhance the efficiency of FLASH: visual token redundancy and vision object co-occurrence. Based on these properties, we design two novel components in the draft model: visual token compression and semi-autoregressive head. Unlike LLMs, LMMs often process a large number of redundant visual tokens, which significantly slow down inference during speculative decoding. To mitigate this, we compress the visual tokens based on the hidden state features, which accelerates draft generation while minimizing the loss of semantic information. Speculative decoding speeds up autoregressive generation using a lightweight draft model to predict the output of a heavier target model. However, since the draft model remains autoregressive, the overall speed-up is limited. In contrast to textual input, visual input inherently exhibits spatial co-occurrence rather than left-to-right causal relationships. Image patches are arranged to reflect their spatial positions, not sequential dependencies. As a result, when describing multiple visual regions simultaneously, models often rely on fixed collocation patterns such as "in front of" or "on the table", which do not require strict autoregressive ordering. This observation motivates our adoption of a semi-autoregressive decoding strategy, which better preserves spatial relationships while maintaining generation efficiency.

By combining latent-aware visual token compression with semi-autoregressive decoding, FLASH achieves faster inference and maintains high draft quality. We evaluate FLASH on video captioning and visual instruction tuning tasks, using LLaVA (Shang et al., 2024) and QwenVL (Bai et al., 2025) as target models. Experimental results demonstrate that the two components of FLASH, visual token compression and semi-autoregressive decoding, provide distinct yet synergistic advantages on different tasks. In video captioning, where a large number of visual tokens are involved, visual token compression is particularly effective, while in instruction tuning, which typically involve fewer visual tokens but longer textual inputs, the semi-autoregressive decoding contributes more significantly to efficiency gains. Overall, FLASH achieves average speed-up gains of $24.4\%$ ($0.41\times$) and $41.5\%$ ($0.62\times$) on video captioning and visual instruction tuning tasks, compared to the previous methods that rely solely on text tokens in multimodal speculative decoding.

We make the following contributions:
(1) We propose FLASH, a novel speculative decoding framework for LMMs that effectively exploits the characteristics of multimodal inputs.
(2) By observing that visual information is often redundant and descriptions of visual content typically appear as short phrases, we introduce visual token compression and semi-autoregressive generation to accelerate draft inference.
(3) Extensive experiments demonstrate that FLASH achieves substantial acceleration in LMMs without notable degradation in draft quality.

## 2 RELATED WORKS

**Speculative decoding.** Since the introduction of speculative sampling (Leviathan et al., 2023; Chen et al., 2023), the strategy of using light-weight models to generate drafts, with large models

performing parallel verification, has been widely adopted to accelerate inference across various LLMs (Xia et al., 2024b; Gao et al., 2025). However, selecting an appropriate draft model is challenging, as it is difficult to make the predicted distribution consistent with that of the target model due to differences in model size or architecture (Bachmann et al., 2025). It has been suggested that knowledge distillation applied to the target model can produce a compact model with a higher reception rate (Zhou et al., 2023). Unlike training a draft model independently, self-speculative decoding introduces a way to reuse components of the target model (Liu et al., 2024a; Elhoushi et al., 2024; Xia et al., 2024a). Following this thought, Eagle (Li et al., 2024b;c) introduces an autoregressive head on the second-to-top feature extracted by the target model to predict candidate tokens. Beyond increasing the acceptance rate of draft tokens, enhancing the drafting efficiency further contributes to achieving a higher speed-up ratio in speculative decoding. Medusa applies n-head architecture, where each head predicts one corresponding token (Cai et al., 2024). Lookahead Decoding leverages an n-gram pool generated through Jacobi iterations, enabling the model to accept multi-token prefixes (Fu et al., 2024). Speculative decoding has also been extended beyond LLMs to LMMs Ganesan et al. (2025); Hu et al. (2025); Wang et al. (2025). MASSV (Ganesan et al., 2025) adapts speculative decoding to multimodal settings via self-distillation, and Dream (Hu et al., 2025) enhances draft quality through a specialized cross-attention mechanism on visual and textual features. Despite these advances, applying speculative decoding to LMMs introduces a fundamental challenge: how to effectively integrate both visual and textual modalities while preserving efficiency and accuracy during inference.

**Token Compression.** Prior studies have shown that incorporating additional inputs can help correct erroneous outputs and enhance model performance (Liu et al., 2025). However, as Large Multimodal Models (LMMs) continue to scale, improving inference efficiency has become increasingly important (Liu et al., 2023; 2024b; Team et al., 2024). A key challenge lies in the high computational cost caused by the large number of visual tokens. To address this, reducing the number of input tokens without sacrificing essential information is considered a crucial strategy. To reduce redundancy, vision tokens can be merged based on their similarity (Shang et al., 2024; Li et al., 2024a). Image token reduction can also be achieved through a Q-Former (Li et al., 2023) to extract of visual concepts (Yang et al., 2024; Chen et al., 2024b). However, evidence from recent research indicates that Q-Former leads to some degree of visual information loss (Yao et al., 2024; Fan et al., 2024). Based on the observation that early-layer visual tokens contain more critical information, LLaVA-mini integrates visual features into text tokens through pre-fusion before these layers (Zhang et al., 2025). DART prioritizes the removal of duplicate tokens over the selection of important tokens (Wen et al., 2025).

## 3 FLASH

### 3.1 PRELIMINARIES

In this paper, our Large Multimodal Models (LMMs) process two data modalities, images and texts. As shown in Figure 1, the image is encoded by a vision encoder and projected into visual embeddings $V = \{V_1, V_2..., V_N\}$, while the text is encoded into textual embeddings $E = \{E_{N+1}, E_{N+2}, ..., E_M\}$, where $N$ denotes the number of visual tokens, and $M$ represents the total number of input tokens, encompassing both visual and textual components. Subsequently, these embeddings are concatenated and processed through Transformer layers to produce the second-to-top-layer feature $F = \{F_1, F_2, ..., F_M\}$. Finally, an LM head maps the last feature $F_M$ to a probability distribution over the output vocabulary space, from which the next token $T_0$ is sampled.

In speculative decoding, the LMM acts as the target model for output validation, while a lightweight draft model proposes $K$ candidate tokens $T'_1, ..., T'_K$. During validation, the target model computes the probabilities of these $K$ candidate tokens in parallel within a single forward pass. The acceptance probability of each candidate token is defined as the ratio of its probability under the target model to its probability under the draft model. Subsequently, the tokens are evaluated after obtaining the acceptance probability of each candidate token. For the $i$-th token $T'_i$, if accepted, it is retained as part of the output. If rejected, the token and all subsequent candidate tokens, $T'_i$ to $T'_K$ are discarded, and the draft model is triggered to generate the next $K$ tokens starting from the position $i$.

Given the single-step inference time $\mathcal{M}_T$ and $\mathcal{M}_D$ for the target and draft model respectively, draft model's inference time $\mathcal{M}_D$ is much less than that of the target model $\mathcal{M}_T$, due to the lightweight architecture of the draft model. Besides, the verification of $K$ candidate tokens by the target model

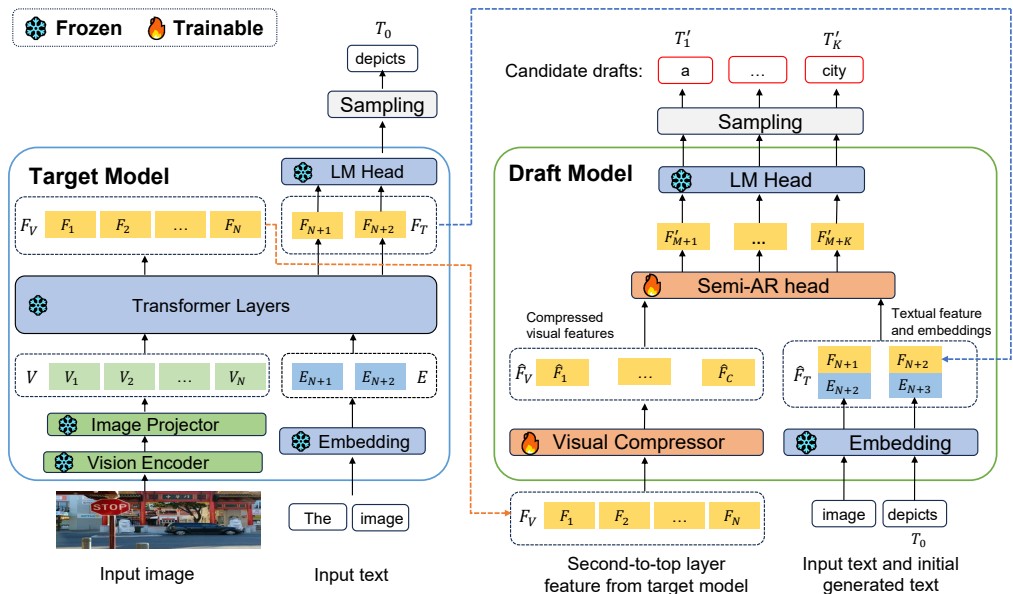

Figure 1: **Illustration of FLASH.** The target model is depicted on the left, while the draft model is shown on the right. In this example, the number of the total input tokens $M$ is equal to $N + 2$, as the input includes $N$ visual tokens and 2 textual tokens. The draft model takes the second-to-top layer features $F_V$ as the visual input. These features are first compressed by a visual compressor, producing the compressed visual features $\hat{F}_V$. Along with textual feature and embeddings $\hat{F}_T$, they are fed into a semi-autoregressive head to generate the next $K$ tokens in parallel. These candidate drafts are highlighted with red-bordered boxes.

is performed in parallel, so the time required to verify all $K$ tokens can be approximated by the single-step time $\mathcal{M}_T$. For speculative decoding, the total time includes the autoregressive draft model generating $K$ tokens and the subsequent verification by the target model, which can be expressed as:

$$\mathcal{M}_T + K \cdot \mathcal{M}_D. \tag{1}$$

When generating $i$ tokens, i.e., the first $i$ tokens of the $K$ candidate tokens are accepted, speculative decoding takes time of $\mathcal{M}_T + K \cdot \mathcal{M}_D$, while standard vanilla autoregressive decoding takes time of $i \cdot \mathcal{M}_T$, since the latter generates $i$ tokens by performing the forward pass $i$ times with the target model. Therefore, the speed-up ratio of autoregressive speculative decoding can be calculated as:

$$\mathcal{R}_{\text{SD}} = \frac{i \cdot \mathcal{M}_T}{(\mathcal{M}_T + K \cdot \mathcal{M}_D)}. \tag{2}$$

When the target model is selected, $\mathcal{M}_T$ remains constant. Therefore, to improve the speed-up ratio, it is crucial to reduce the draft generation time $K \cdot \mathcal{M}_D$, while maintaining a high draft quality to ensure a large value of $i$. To achieve this, we propose a novel method named FLASH, which effectively balances inference speed and draft quality by leveraging the characteristics of multimodal inputs. Following the previous LLM speculative decoding method Eagle (Li et al., 2024b), we construct the textual input $\hat{F}_T$ by concatenating the second-to-top layer textual feature $F_T$ and the token embeddings $E$. For the visual modality, FLASH introduces a visual token compression module that reduces computational overhead while preserving key visual semantics. To further accelerate decoding, FLASH incorporates a semi-autoregressive head that predicts the next $K$ tokens in parallel, significantly reducing the draft generation time from $K \cdot \mathcal{M}_D$ to approximately $\mathcal{M}_D$.

## 3.2 VISUAL TOKEN COMPRESSION

Differ from LLM speculative decoding methods, our input consists of both visual and textual tokens. Compared to textual tokens, visual tokens are often numerous, leading to a significant increase in inference time (Zhang et al., 2025). To balance the acceleration of draft generation and the potential

degradation on acceptance rate, we propose an effective strategy for compressing the visual tokens in the draft model, allowing for faster inference without significantly compromising prediction quality. Since some visual tokens contain redundant information and do not significantly contribute to the final prediction, our goal is to compress the $N$ visual tokens into a smaller set of size $C$. Specifically, the $N$-sized visual feature $F_V$, corresponding to the $N$ visual tokens, is fed into a visual compressor to produce a compressed feature $\hat{F}_V$ of size $C$. Inspired by the attention mechanism, we introduce a learnable query set $\mathcal{C}$ of size $C$, where each query is designed to extract information from the feature $F_V$. By learning a compressed feature representation of size $C$, the model is able to retain the most salient semantics from the original visual feature. Specifically, the compressed feature $\hat{F}_V$ is computed as:

$$\hat{F}_V = \text{softmax}(\mathcal{C} \cdot F_V^T) \cdot F_V, \tag{3}$$

where $\mathcal{C}$ serves as the query, and visual feature $F_V$ acts as both the key and value in the attention computation. The softmax operation normalizes the attention scores, which are then used to aggregate $F_V$ into the compressed representation. Eventually, the scale of the visual inputs for the draft model is reduced from $N$ to $C$.

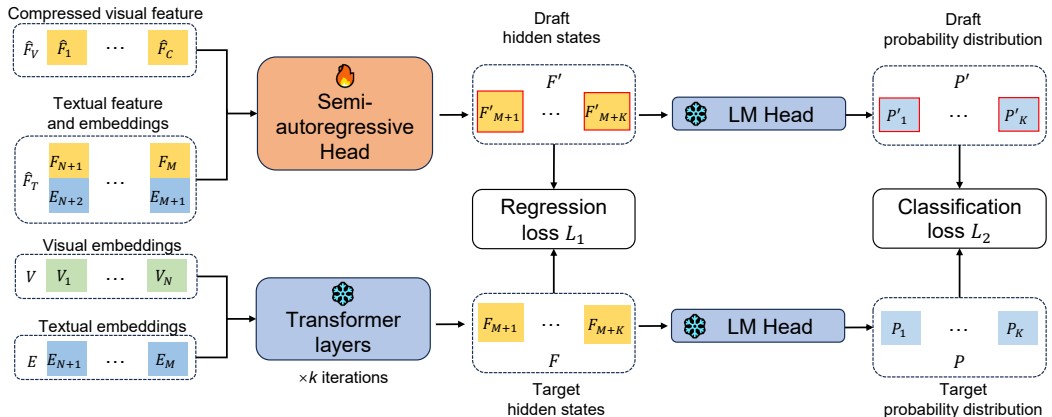

Figure 2: **Training procedure with the semi-autoregressive head.** The semi-autoregressive head concatenates the compressed visual features $\hat{F}_V$ and the textual features and embeddings $\hat{F}_T$ as input, and subsequently generates the hidden state features $F'$ for the next $K$ tokens. The corresponding probability $P$ for these $K$ candidate tokens are calculated by the frozen LM head from the target model. The semi-autoregressive head is trained using a regression loss, where the feature $F$ from the frozen target model is served as the ground truth, as well as a classification loss, where the draft probability distribution $P'$ is supervised by the output distribution of the target model $P$.

### 3.3 SEMI-AUTOREGRESSIVE INFERENCE

In contrast to the previous speculative decoding methods, we introduce a semi-autoregressive head to predict the next $K$ tokens in parallel. Specifically, when generating $K$ candidate tokens, unlike an autoregressive draft model with a runtime of $K \cdot \mathcal{M}_D$ time for performing $K$ inference steps, our semi-autoregressive approach needs only a single inference pass of the draft model, with a runtime of approximately $\mathcal{M}_D$.

To enable parallel speculative decoding, we feed the draft model with $K$ placeholder tokens. The model then produces predictions for the next $K$ positions. As Figure 2 shows, the semi-autoregressive head outputs hidden states $F'_{M+1:M+K}$, which are projected by the frozen LM head into probability distributions $P'_{1:K}$, from which candidate tokens $T'_{1:K}$ are sampled.

During training, our loss function comprises two components. The first is a regression loss, where we employ the Smooth L1 loss to quantify the difference between the features predicted by the draft model $F'_{M+1:M+K}$ and those from the target model $F_{M+1:M+K}$. The second component is a classification loss, where we employ the cross-entropy loss to encourage the draft model to generate

tokens that align with those produced by the target model. The total loss is defined as:

$$L = \sum_{i=M+1}^{M+K} \ell_{\text{reg}}(F_i', F_i) + \alpha \cdot \sum_{i=1}^{K} \ell_{\text{cls}}(P_i', P_i), \qquad (4)$$

where $\ell_{\text{reg}}(\cdot, \cdot)$ and $\ell_{\text{cls}}(\cdot, \cdot)$ denotes the Smooth L1 loss and the cross-entropy loss, respectively. $\alpha$ is a hyper-parameter used to balance the regression and classification losses, ensuring they are of comparable magnitudes. Specifically, $\alpha$ is set to $0.1$. After computing the loss at each step, the next token predicted by the target model is appended to the input sequence. The extended sequence is then used to predict the subsequent $K$ tokens. This iterative procedure continues until the entire output sequence is generated.

During inference, the draft model produces $K$ candidate tokens in a single forward pass. These candidate tokens are then sequentially verified by the target model. The draft generation process is outlined in Algorithm 1 (in Supplementary Materials).

## 4 EXPERIMENTS

### 4.1 IMPLEMENTATION SETTINGS

To evaluate the effectiveness of FLASH, we conduct experiments on LLaVA-1.5 and QwenVL-2.5 models. We train and evaluate on video captioning and visual instruction tuning task, using Kinetics-400 (Kay et al., 2017) datasets and LLaVA-instruct-150k (Liu et al., 2023), respectively. Specifically, Kinetics-400 is a video dataset containing 400 action classes. Since it only provides category annotations, we use the corresponding target model to generate captions as pseudo ground truth. LLaVA-Instruct-150K consists of multimodal instruction-response pairs generated by GPT. For both tasks, we sample 10k instances from each dataset for training. The maximum sequence lengths of training data are set to 200 for video captioning and 2048 for visual instruction tuning.

The learning rate is set to $2 \times 10^{-5}$ and the batch size is set to 4. To ensure a fair comparison, all inferences are performed on a single NVIDIA A6000 GPU. During the evaluation, the batch size is set to 1, following standard practice in prior LLM speculative decoding methods (Li et al., 2024b; Chen et al., 2024c).

We introduce two metrics to evaluate the effectiveness of the speculative decoding: the average acceptance tokens $\mathcal{A}$ and the speed-up ratio $\mathcal{R}$. The average acceptance tokens $\mathcal{A}$ is defined as the average number of the accepted tokens during a single forward pass of the draft model. Since multiple candidate tokens are generated in parallel during semi-autoregressive inference, the average acceptance tokens $\mathcal{A}$ can be greater than 1, which distinguishes it from autoregressive speculative decoding methods. The speed-up ratio $\mathcal{R}$ is defined as the generation speed of the target model divided by the total time required for token generation and verification under speculative decoding. We adopt speed-up ratio $\mathcal{R}$ as the primary metric to assess the overall efficiency.

### 4.2 RESULTS

We perform experiments on two typical multimodal tasks: video captioning, which involves a large number of visual tokens as input, and visual instruction tuning, which processes a single image accompanied by textual interactions. Next, we analyze the experimental results for each task individually.

#### 4.2.1 VIDEO CAPTIONING

Table 1 shows the results when using LLaVA-1.5 and QwenVL-2.5 as the target model. $\tau$ represents the temperature of the target model, with $\tau = 0$ corresponding to greedy decoding and $\tau = 1$ leading to more diverse outputs. For a fair comparison, the candidate draft length $K$ is fixed at 4 across all speculative decoding models.

In video captioning, the primary challenge lies in effectively disposing of the visual content while ensuring that the draft model can generate accurate captions. Among the competing models, "Text-only" refers to a speculative decoding method which only considers the textual input, following the

Table 1: Quantitative experiments of **video captioning** on LLaVA-1.5 and QwenVL-2.5. We report the results of speed-up ratio $\mathcal{R}$ and the average acceptance tokens $\mathcal{A}$ under temperature $\tau = 0$ and $\tau = 1$. Additionally, we include the average computational cost (FLOPs), for a single round of draft generation and validation. FLASH combines "VisComp" and "SemiAR", achieving the best speed-up ratio among all competing methods, while maintaining output consistency with the target model. Underline indicates the best result, while green denotes the second-best result.

| LLaVA-1.5 | $\mathcal{A}\uparrow$ | $\mathcal{R}\uparrow$ | LLaVA-1.5 | $\mathcal{A}\uparrow$ | $\mathcal{R}\uparrow$ | FLOPs$\downarrow$ |
|---|---|---|---|---|---|---|
| Speculative Decoding | | | Speculative Decoding | | | |
| Text-only | 0.59 | 1.42× | Text-only | 0.52 | 1.37× | 70.4T |
| Eagle-MM | 0.69 | 1.63× | Eagle-MM | 0.66 | 1.60× | 79.5T |
| Medusa-MM | 2.31 | 1.69× | Medusa-MM | 2.27 | 1.63× | 80.2T |
| $\tau$=0  Dream | 0.70 | 1.66× | $\tau$=1  Dream | 0.68 | 1.63× | 79.9T |
| Ours | | | Ours | | | |
| VisComp | 0.68 | 1.79× | VisComp | 0.66 | 1.70× | 71.2T |
| SemiAR | 2.65 | 1.77× | SemiAR | 2.64 | 1.76× | 74.5T |
| FLASH | 2.63 | 1.83× | FLASH | 2.63 | 1.81× | 70.7T |
| QwenVL-2.5 | $\mathcal{A}\uparrow$ | $\mathcal{R}\uparrow$ | QwenVL-2.5 | $\mathcal{A}\uparrow$ | $\mathcal{R}\uparrow$ | FLOPs$\downarrow$ |
| Speculative Decoding | | | Speculative Decoding | | | |
| Text-only | 0.70 | 2.33× | Text-only | 0.54 | 1.61× | 56.3T |
| Eagle-MM | 0.83 | 2.49× | Eagle-MM | 0.80 | 1.93× | 65.5T |
| Medusa-MM | 2.76 | 2.41× | Medusa-MM | 2.65 | 2.35× | 67.3T |
| $\tau$=0  Dream | 0.83 | 2.51× | $\tau$=1  Dream | 0.82 | 2.00× | 65.7T |
| Ours | | | Ours | | | |
| VisComp | 0.83 | 2.60× | VisComp | 0.79 | 1.99× | 57.3T |
| SemiAR | 3.28 | 2.63× | SemiAR | 3.09 | 2.00× | 61.6T |
| FLASH | 3.21 | 2.68× | FLASH | 2.98 | 2.05× | 56.8T |

approach proposed by (Gagrani et al., 2024). "Eagle-MM" and "Medusa-MM" refer to variants of Eagle (Li et al., 2024b) and Medusa (Cai et al., 2024), to process image inputs by incorporating a CLIP-style visual encoder and projector, which align visual and textual embeddings into a shared space. These aligned embeddings are concatenated to form the input embeddings and are combined with the second-to-top layer feature $F$, following the same design as used in the textual components. We observe that although text-only speculative decoding offers some acceleration, its speed-up ratio is inferior to that of multimodal speculative decoding. This is primarily because relying solely on textual context often causes the draft model to generate outputs that diverge from the actual visual content, thereby reducing the overall quality of the generated drafts.

In the LLaVA model, where each image initially consumes 576 tokens, our visual token compression method reduces this to 64 tokens. Comparing previous speculative decoding method "Eagle-MM" and "Medusa-MM" with "VisComp", we observe that incorporating vision token compression significantly reduces the computational load (FLOPs) and improves the speed-up ratio $\mathcal{R}$ by reducing the number of input tokens, which causes only a minimal decrease in the average acceptance tokens $\mathcal{A}$. Compared to speculative decoding in the autoregressive setting ("Eagle-MM"), the semi-autoregressive approach ("SemiAR") demonstrates a significant advantage in terms of average accepted tokens, as it can return multiple draft candidates in a single forward pass. This leads to an overall speed-up ratio improvement of approximately 9.3%. By combining visual token compression with semi-autoregressive inference, FLASH achieves the highest speed-up ratio at about $1.8\times$ compared to the target model.

In the QwenVL model, the number of visual tokens varies with the input image size. To standardize, we resize images to generate 324 tokens per image. Additionally, our visual token compression reduces the token count to 36. Most trends in the QwenVL model align with those observed in LLaVA. Compared to "Eagle-MM", "VisComp" and "SemiAR" yield additional speed-up gains of approximately 2.6% and 6.3%, respectively. Furthermore, FLASH achieves speed-up ratios of $2.68\times$ and $2.05\times$ over target model under $\tau = 0$ and $\tau = 1$, respectively. These results demonstrate the effectiveness of our method on both LLaVA and QwenVL models.

Table 2: Quantitative experiments of **visual instruction tuning** on LLaVA-1.5 and QwenVL-2.5. Similar to video captioning task, we report speed-up ratio $\mathcal{R}$, average acceptance tokens $\mathcal{A}$, and average computational cost (FLOPs). FLASH combines "VisComp" and "SemiAR", achieving the highest speed-up ratio among all competing methods, while maintaining output consistency with the target model. Underline indicates the best result, while green denotes the second-best result.

| LLaVA-1.5 | $\mathcal{A}\uparrow$ | $\mathcal{R}\uparrow$ | LLaVA-1.5 | $\mathcal{A}\uparrow$ | $\mathcal{R}\uparrow$ | FLOPs$\downarrow$ |
|---|---|---|---|---|---|---|
| Speculative Decoding | | | Speculative Decoding | | | |
| Text-only | 0.68 | 1.60× | Text-only | 0.66 | 1.52× | 11.6T |
| Eagle-MM | 0.76 | 2.21× | Eagle-MM | 0.76 | 2.19× | 12.7T |
| Medusa-MM | 2.55 | 2.28× | Medusa-MM | 2.52 | 2.24× | 13.0T |
| $\tau=0$   Dream | 0.72 | 2.18× | $\tau=1$   Dream | 0.70 | 2.14× | 12.9T |
| Ours | | | Ours | | | |
| VisComp | 0.76 | 2.23× | VisComp | 0.76 | 2.21× | 11.8T |
| SemiAR | 2.86 | 2.49× | SemiAR | 2.58 | 2.24× | 12.0T |
| FLASH | 2.77 | 2.55× | FLASH | 2.59 | 2.29× | 11.5T |
| QwenVL-2.5 | $\mathcal{A}\uparrow$ | $\mathcal{R}\uparrow$ | QwenVL-2.5 | $\mathcal{A}\uparrow$ | $\mathcal{R}\uparrow$ | FLOPs$\downarrow$ |
| Speculative Decoding | | | Speculative Decoding | | | |
| Text-only | 0.52 | 1.44× | Text-only | 0.50 | 1.36× | 11.4T |
| Eagle-MM | 0.65 | 1.63× | Eagle-MM | 0.63 | 1.59× | 12.6T |
| Medusa-MM | 2.36 | 1.68× | Medusa-MM | 2.29 | 1.65× | 13.0T |
| $\tau=0$   Dream | 0.67 | 1.66× | $\tau=1$   Dream | 0.66 | 1.65× | 12.7T |
| Ours | | | Ours | | | |
| VisComp | 0.63 | 1.69× | VisComp | 0.60 | 1.65× | 11.6T |
| SemiAR | 2.46 | 1.79× | SemiAR | 2.38 | 1.70× | 11.8T |
| FLASH | 2.46 | 1.83× | FLASH | 2.36 | 1.71× | 11.3T |

### 4.2.2 VISUAL INSTRUCTION TUNING

In visual instruction tuning, each input consists of an image paired with a textual instruction, covering a broad range of tasks such as image captioning, visual question answering, object recognition, and visual reasoning. Using LLaVA-1.5 and QwenVL-2.5 as the target models, we conduct the instruction tuning experiments following the same experiment setups as in the video captioning task.

As presented in Table 2, by comparing the "Text-only" with "Eagle-MM" speculative decoding, we find that relying solely on text input leads to a significant drop in the draft generation quality when fine-grained image understanding is required, particularly on LLaVA. Additionally, our proposed visual token compression ("VisComp") and semi-autoregressive ("SemiAR") brings overall 2.3% and 7.9% speed-up ratio improvements, comparing to "Eagle-MM". Specifically, we further find that when temperature $\tau = 0$, using "SemiAR" yields a 0.22× speed-up, which is notably higher than the 0.08× improvement observed at $\tau = 1$. This improvement due to the model's tendency to produce more deterministic outputs at lower temperatures, which aligns well with the parallel token generation in the semi-autoregressive head.

Notably, "SemiAR" yields greater improvements in visual instruction tuning than in video captioning. Besides, we observe that the improvement brought by visual token compression is less pronounced in the visual instruction tuning task compared to the video captioning task. This is likely because visual instruction tuning typically involves a single image, whereas video captioning tasks process multiple frames. By combining "VisComp" and "SemiAR", FLASH achieves speed-up ratios of 2.55× and 1.83× on LLaVA and QwenVL, respectively. Moreover, we observe that FLASH efficiently processes visual information while incurring a computational load comparable to that of the "Text-only" model. It enables FLASH to achieve a significantly higher speed-up ratio without sacrificing draft quality. These results further demonstrate the effectiveness of our method across both video captioning and visual instruction tuning tasks.

### 4.2.3 TEXT-ONLY SCENARIO

To further investigate the effectiveness of our method beyond multimodal tasks, we additionally evaluate FLASH in text-only setting. In this case, the visual token compression module becomes inactive, and the draft model operates solely based on the textual prompt.

Table 3: Speed-up ratios under text-only prompts on LLaVA-650k subset.

| Method | Image-text pairs | Text-only |
|---|---|---|
| Text-only | - | $1.63\times$ |
| Eagle-MM | $2.21\times$ | $1.82\times$ |
| FLASH | $2.49\times$ | $1.88\times$ |

The results on LLaVA-650k text-only subset are summarized in Table 3. Several observations can be drawn. First, speculative decoding in a text-only setting achieves a $1.63\times$ speed-up, which is lower than that in the multimodal setting. The results indicate that visual input can provide structural cues (e.g., spatial relations) that tend to make generation more predictable. In comparison, text-only sequences generally display more diverse linguistic patterns. Second, FLASH attains $1.88\times$ speed-up in the text-only case ($+0.25\times$ over the text-only baseline), and $2.49\times$ in the multimodal setting ($+0.28\times$ over Eagle-MM).

Overall, these results confirm two key insights: First, visual grounding indeed provides additional structural constraints that amplify the benefits of speculative decoding. Second, FLASH's semi-autoregressive design and draft architecture remain effective in pure-text scenarios, even though the model is primarily trained on image–text pairs. We hypothesize that explicit exposure to text-only data during training could further boost performance in such cases. These findings highlight the distinct structural properties of multimodal generation and reinforce our claim that semi-autoregressive decoding is particularly well-suited to vision–language models.

### 4.3 ABLATIONS

To further clarify the contribution of the visual token compression module, we perform an ablation where FLASH is evaluated with different visual compression methods. Results are shown in Table 4.

Table 4: Ablation study on different visual token compression methods.

| Method | Visual Instruct Tuning | Video Captioning |
|---|---|---|
| w/o VisComp | $2.49\times$ | $1.77\times$ |
| Average pooling | $2.31\times$ | $1.77\times$ |
| SGL (Zhao et al., 2025) | $1.68\times$ | $1.80\times$ |
| FastV (Chen et al., 2024a) | $1.53\times$ | $1.60\times$ |
| Twig (Shao et al., 2025) | $1.58\times$ | $1.53\times$ |
| FLASH (ours) | $2.55\times$ | $1.83\times$ |

We observe that our FLASH achieves the best acceleration consistently across both tasks ($2.55\times$ and $1.83\times$). Non-parametric strategies such as average pooling and SGL Zhao et al. (2025) yield limited speed-up ratio. It suggests that visual token compression module goes beyond simple redundancy-reduction heuristics, providing a more principled design to maximize efficiency. In contrast to previous visual token compression methods like FastV Chen et al. (2024a) and Twig Shao et al. (2025), which primarily aim to shorten the input sequence and preserve semantic information, our approach emphasizes inference efficiency. As a result, while their acceleration is limited, it reflects the fact that their design focus was not faster decoding but semantic preservation under compression.

We further conduct ablations on the scale of $\mathcal{C}$, which influences the number of visual tokens after compression. The results are presented in Table 5. Our chosen ratio of $1/9$ strikes an optimal balance, delivering the peak speed-up. The performance remains robust across a range of ratios (from $1/4$ to $1/16$), indicating that FLASH is not overly sensitive to the exact compression rate. However, the extreme case of removing visual tokens entirely ($C/N = 0$) leads to a clear performance drop, confirming that our compression module retains essential visual information necessary for maintaining draft quality.

Table 5: Speed-up Ratios with different compression ratios on visual instruct tuning and video captioning.

| Compression ratios C/N | Visual Instruct Tuning | Video Captioning |
|---|---|---|
| 1 (w/o compression) | 2.49× | 1.77× |
| 1/4 | 2.51× | 1.83× |
| 1/9 (FLASH) | 2.55× | 1.83× |
| 1/16 | 2.33× | 1.80× |
| 0 (w/o image) | 1.92× | 1.60× |

## 4.4 SCALE OF DRAFT MODEL

To further explore the trade-off between the draft quality and its efficiency, we vary the scale of the draft model through two approaches: adjusting the number of parameters and applying quantization. First, we modify the number of parameters in the draft model, including the number of attention heads and the number of attention layers. The resulting speed-up ratios are reported in Table 6. These

Table 6: Speed-up ratios under different draft model scales.

| Task | Visual Instruct Tuning | Video Captioning |
|---|---|---|
| Head-32/Layer-1 (FLASH) | 2.55× | 1.83× |
| Head-64/Layer-1 | 2.56× | 1.80× |
| Head-16/Layer-1 | 2.11× | 1.34× |
| Head-32/Layer-2 | 1.97× | 1.55× |

results show that balance between the scale of draft model and speed is important. For example, a draft model with 2 Transformer layers may have more accuracy in predicting the target distributions. However, at the same time, it introduces additional computational cost thus affect the overall speed-up ratios.

Second, we conducted additional experiments by applying quantization to the draft model. The measured speed-up ratios are summarized in Table 7.

Table 7: Quantization on draft model, reporting the speed-up ratio as the primary metric.

| Task | Visual Instruct Tuning | Video Captioning |
|---|---|---|
| FLASH-fp16 | 2.55× | 1.83× |
| FLASH-bf16 | 2.43× | 1.79× |
| FLASH-int8 | 2.44× | 1.74× |
| FLASH-int4 | 2.15× | 1.55× |

These results show that quantization does not offer the expected re-speeding up. We believe this is because our draft model is already lightweight. In such a small model, the efficiency gains from quantization are insufficient to offset the loss in prediction accuracy. We will incorporate an explicit discussion of these trade-offs in the revised manuscript, as we agree it is an important consideration for the broader speculative decoding paradigm and its practical applications.

## 5 CONCLUSIONS

In this work, we introduce FLASH, a novel speculative decoding framework designed specifically for Large Multimodal Models (LMMs). FLASH introduces two key components: a visual token compression module that reduces input redundancy, and a semi-autoregressive decoding strategy that enables parallel token generation. Together, these innovations accelerate multimodal inference while preserving the quality of the output. Experiments on video captioning and visual instruction tuning tasks demonstrate that FLASH delivers substantial speed-ups over existing speculative decoding methods, enabling more efficient deployment of LMMs.

## 6 ETHICS STATEMENT

This work adheres to the ICLR Code of Ethics . Our research does not involve human subjects, sensitive personal data, or personally identifiable information. No new datasets containing private or ethically sensitive content were collected, and all datasets used in this paper are publicly available and widely adopted in the research community. The study does not present any foreseeable risks of harmful applications, discrimination, or privacy/security concerns.

## 7 REPRODUCIBILITY STATEMENT

Our code is available via the [anonymous link]. The experiment settings, including training and evaluating details, are provided in Section 4.1. All datasets used in our experiments are publicly available.

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

# A APPENDIX

## A.1 USE OF LARGE LANGUAGE MODELS

We used large language models (LLMs) solely for translation and language polishing of the manuscript. They were not involved in designing methods, conducting experiments, or analyzing results.

## A.2 ALGORITHM

Algorithm 1 illustrates the draft generation process.

---

**Algorithm 1** Draft Generation Process

---

**Input:** Visual embeddings $V$, Textual embeddings $E$, Number of tokens $K$, Semi-autoregressive head $S$, Compression query $\mathcal{C}$.

**Output:** Generated draft tokens $T'_1, ..., T'_K$.

**Step 1: Transformer Processing**

1: $F_V, F_T \leftarrow$ Transformer_layers(concat($V, E$)) // Second-to-top layer visual and textual features

**Step 2: Visual token Compression**

2: $\hat{F}_V \leftarrow$ softmax $\left(\mathcal{C} \cdot F_V^T\right) \cdot F_V$             // Compressed features

**Step 3: Semi-autoregressive Generation**

3: $\hat{F}_T \leftarrow$ FC(concat($F_T, E_{N+2:M+1}$))             // Textual input

4: $F' \leftarrow$ concat($\hat{F}_V, \hat{F}_T$)             // Concatenate visual and textual input

5: $F'_{M+1}, ..., F'_{M+K} \leftarrow S(F')$             // Semi-autoregressive inference

6: $P'_1, ..., P'_K \leftarrow$ LM-head($F'_{M+1}$), ..., LM-head($F'_{M+K}$)      // Probability distribution of $T'_i$

7: **return** $T'_1, ..., T'_K \leftarrow$ argmax($P'_1$), ..., argmax($P'_K$)

---

### A.3 ABLATION ON K

In traditional speculative decoding, the draft model generates tokens sequentially in an autoregressive manner over $K$ steps. The hyperparameter $K$ is utilized during inference to determine the number of speculative tokens generated per iteration; however, it is not involved in the training phase of the draft model. In contrast, our method employs a semi-autoregressive head to generate $K$ tokens in parallel, making $K$ a critical hyperparameter during both training and inference.

Figure 3 illustrates the impact of different $K$ on the speed-up ratio and average acceptance tokens. When $K = 1$, the draft model generates a single candidate token per forward pass. This model thereby degenerates into an autoregressive speculative decoding. We observe a notable improvement in both the speed-up ratio and average acceptance tokens when increasing $K$ to 4. However, further increasing $K$ to 6 results in only marginal benefits. The average acceptance tokens $\mathcal{A}$ does not increase proportionally with $K$, suggesting a higher likelihood of token rejection at a larger $K$.

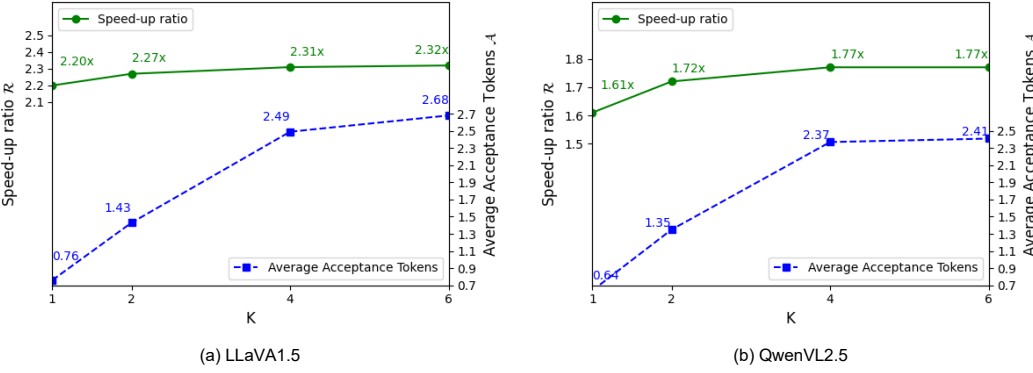

(a) LLaVA1.5             (b) QwenVL2.5

Figure 3: Speed-up ratio and average acceptance tokens with different draft length K. The green solid line indicates the speed-up ratio, while the blue dashed line represents the average acceptance tokens.

Table 8: Quantitative experiments of **video captioning**, reporting the speed-up ratio as the primary metric.

| Model | LLaVA-7B | LLaVA-13B | QwenVL-3B | QwenVL-7B | QwenVL-32B |
|---|---|---|---|---|---|
| Eagle-MM | 1.63× | 1.24× | 2.48× | 2.49× | 2.01× |
| FLASH | 1.83× | 1.47× | 2.65× | 2.68× | 2.20× |

Table 9: Quantitative experiments of **instruction tuning**, reporting the speed-up ratio as the primary metric.

| Model | LLaVA-7B | LLaVA-13B | QwenVL-3B | QwenVL-7B | QwenVL-32B |
|---|---|---|---|---|---|
| Eagle-MM | 2.21× | 1.21× | 1.88× | 1.63× | 1.40× |
| FLASH | 2.55× | 1.49× | 2.01× | 1.83× | 1.49× |

## A.4 SCALABILITY

To validate the robustness of our approach, we further conducted experiments using LLaVA-13B, QwenVL-3B,QwenVL-32B, in addition to the main results reported on LLaVA-7B and QwenVL-7B. The results in Table 8 and Table 9 show that FLASH consistently achieves a higher speed-up ratio compared to multimodal speculative decoding, a variant of Eagle adapted to handle multimodal input (Eagle-MM). This highlights the efficiency of FLASH in both video captioning and instruction tuning tasks.

## A.5 MULTI-TASK EVALUATION OF VISUAL INSTRUCTION TUNING

Visual instruction tuning encompasses multiple tasks, such as fine-grained image captioning, ScienceQA, and commonsense reasoning. We evaluate the performance on each benchmark individually. The results demonstrate that FLASH consistently outperforms the competing methods across all tasks, indicating its robustness.

Table 10: Performance comparison on multiple tasks.

| Method | COCO Captioning | MMT-Bench | ScienceQA |
|---|---|---|---|
| EAGLE-MM | 2.28× | 2.26× | 2.15× |
| DREAM | 2.49× | 2.23× | 2.11× |
| FLASH | 2.64× | 2.38× | 2.32× |

## A.6 LIMITATIONS AND FUTURE WORKS

Experiments demonstrate that FLASH consistently accelerates both video captioning and visual instruction tuning. Importantly, FLASH has the potential to serve as a general-purpose acceleration method on a wide range of multimodal tasks, as visual instruction tuning itself encompasses multiple such tasks. We intend to extend our approach to additional multimodal tasks in future work. Furthermore, the semi-autoregressive decoding strategy can experience reduced acceptance rates as the draft length increases. Although these trade-offs did not significantly impact the results reported here, they indicate that further investigation into adaptive compression techniques and more flexible decoding strategies will be important for achieving broader and more reliable deployment.

