# OpenReview forum: "FLASH: Latent-Aware Semi-Autoregressive Speculative Decoding for Multimodal Tasks"
_ICLR.cc/2026/Conference — Submitted to ICLR 2026_

### Official Review · Reviewer_ht7i · 2025-10-25

**Soundness:** 2
**Presentation:** 3
**Contribution:** 2
**Rating:** 4
**Confidence:** 4

**Summary:**

This paper proposes FLASH, a speculative decoding method for LMMs. The authors introduce two key components in the draft model: visual token compression to reduce redundant visual tokens, and a semi-autoregressive head to exploit spatial co-occurrence patterns in visual inputs. FLASH outperforms existing speculative decoding methods for LMMs.

**Strengths:**

1. The paper addresses an important and timely problem in speculative decoding for LMMs.
2. The paper is well written and easy to follow.

**Weaknesses:**

1. The core components (visual token compression and semi-autoregressive generation) are not novel. The main contribution is their effective integration, which is a valuable engineering solution but offers limited algorithmic novelty.

2. The two motivations (visual token redundancy and vision object co-occurrence) of this work are not convincing.

   a. Visual token redundancy: Context token count primarily affects prefill latency, not decode latency. Since speculative decoding targets decode-stage acceleration, why is visual token compression critical here?

   b. Vision object co-occurrence: The claim that visual input uniquely exhibits spatial co-occurrence is unconvincing. Spatial patterns like "in front of" or "on the table" are equally common in text-only contexts, thus the motivation for semi-autoregressive decoding needs stronger justification.

3. While the paper includes the 'Text-only' baseline from Gagrani et al. (2024), it omits Gagrani's multimodal drafting baseline. Furthermore, the paper lacks a comparison with MASSV.

4. Table 1 shows an inconsistency: for QwenVL-2.5 (temperature=1), Medusa-MM's speedup (2.35x) is superior to FLASH's (2.05x), yet the table seems to incorrectly mark FLASH as best. This raises reliability concerns. Furthermore, the speedup from FLASH often appears incremental.

**Questions:**

Please refer to the Weaknesses section above.

---

> ### Author Response · Authors · 2025-11-20
> **Response to Reviewer ht7i**
>
> Thank you for the valuable comments. We add additional ablations to address these points and will integrate the results into the revision. Please let us know if further clarification or new analyses would be helpful.
>
> **Novelty:** FLASH is more than a combination of existing techniques. While speculative decoding itself is a known mechanism, the core contributions of FLASH lie in how speculative decoding is adapted and made effective for vision–language models, which introduces several technical innovations that are not present in prior work. First, unlike previous multimodal speculative decoding methods that rely on average pooling, FLASH introduces a learnable query-based visual compression module that selects the most informative visual features for speculative decoding. This design allows to aggressively reduce visual tokens while preserving semantics, leading to a substantially better accuracy–latency trade-off than all prior compression strategies (Table 4). Second, FLASH proposes a semi-autoregressive draft head that generates multiple draft tokens in parallel while remaining compatible with the target model’s verification behavior. Existing speculative decoding works for VLMs still rely on strictly autoregressive draft generation; our semi-autoregressive design is new, effective, and contributes significantly to the observed acceleration (Tables 1 and 2).
>
> **Visual token redundancy:** We will clarify in the revision why visual token redundancy is directly relevant to speculative decoding efficiency. While it is true that visual tokens mainly affect prefill latency, they also have a substantial impact on the draft model’s decoding cost. During speculative decoding, the computation of the draft model is also memory-bound due to the KV-cache accesses in each decoding step. A large number of visual tokens leads to a large draft-side KV cache, which increases memory traffic and slows down every draft forward pass. By compressing visual tokens, we drastically reduce the KV-cache size for the draft model. This directly improves draft generation speed because of the smaller KV cache and lower attention cost in the draft model. Empirically, we observe that reducing the redundant visual tokens significantly accelerates draft decoding (c.f. Table 1 and Table 2, without visual token compression results in a decrease in speed-up ratios).
>
> **Vision object co-occurrence:** Our claim is not that spatial or relational patterns are unique to vision–language models, but rather that they are significantly more predictable when paired with visual evidence, which directly benefits semi-autoregressive drafting. In text-only generation, relational patterns (e.g., “on the table”, “in front of”) depend purely on linguistic context and may vary widely depending on discourse intent. In contrast, in vision–language models, these patterns are visually anchored: once the model identifies certain objects and their spatial configuration, the corresponding textual expressions become highly constrained and strongly co-occurring. For example, once the image reveals a “cup on the table”, the subsequent tokens describing this relation are far more deterministic than text generation.
> We will revise the paper to more clearly articulate that motivation is not the existence of co-occurrence per se, but the predictability and grounding provided by visual input, which creates a favorable structure for semi-autoregressive speculative decoding.
>
> **Comparison with MASSV and Gagrani et al. (2024):** We will add the comparison with MASSV. Regarding Gagrani et al. (2024), we would like to clarify why we reported it as the "text-only". Although the target model in their setup supports multimodal inputs, the draft model is trained and operated purely on text sequences. It does not consume visual embeddings and does not generate multimodal-conditioned draft predictions. Therefore, the drafting process itself is text-only. For this reason, we categorize it as "text-only".
>
> **Results in Table 1:** We will correct the labeling in Table 1 in the revision.
> Regarding the perception that FLASH’s speedup is incremental, it is important to note that achieving speedup in multimodal settings is generally much harder than in text-only scenarios due to the large number of high-dimensional visual tokens and the complex target model verification. In this context, FLASH is nontrivial and practically significant, representing one of the highest improvements reported in multimodal speculative decoding.
> Moreover, the speculative decoding mechanism guarantees that the generated text strictly adheres to the target model’s distribution. This is fundamentally different from previous compression-based acceleration methods, which often trade off accuracy for speed. Achieving lossless acceleration in multimodal generation is a meaningful contribution, as it enables faster inference without compromising generation quality or model fidelity.

---

> ### Comment · Area_Chair_uUcq · 2025-11-28
>
> Dear Reviewer, the discussion period is about to close. We kindly ask you to participate in the discussion or update your score based on the authors' rebuttal before the deadline. Thank you for your time and valuable contribution!

---

### Official Review · Reviewer_Lox2 · 2025-10-31

**Soundness:** 2
**Presentation:** 3
**Contribution:** 2
**Rating:** 4
**Confidence:** 4

**Summary:**

This work introduces a method to accelerate the inference speed of large vision-language models. The core idea is to leverage a smaller, faster "draft model" to generate a sequence of visual and text tokens as a "draft," which is then efficiently verified in a single parallel pass by the larger, more accurate "target model." By accepting long sequences of drafted tokens when the draft is correct, this speculative decoding approach significantly reduces the number of sequential forward passes required from the expensive target model, thus lowering overall latency.

**Strengths:**

The primary strength of this method lies in its practical utility for reducing the high computational cost associated with large multimodal models. By intelligently combining existing techniques like visual token compression and speculative decoding, it offers a direct path to faster inference without compromising the output quality of the original target model. The ability to verify multiple tokens in parallel is a clever way to exploit the architecture of modern transformers for a substantial speedup.

**Weaknesses:**

The primary weakness is the limited scope of the evaluation. While the reported speedup is promising, the benchmarks used do not cover a diverse range of real-world scenarios. The acceleration factor, presented as an average, may not be representative of performance on tasks with varying levels of predictability, such as complex visual reasoning versus simple image captioning. For instance, the actual performance gain in applications requiring high creativity or nuanced understanding could be significantly lower than reported. Furthermore, the novelty is somewhat constrained, as the approach is fundamentally an application and combination of pre-existing techniques rather than the introduction of a completely new paradigm.

**Questions:**

The methodology for creating the draft model raises a key question. Is the draft model a fine-tuned version of the same architecture as the target model? If so, have the authors considered replacing its backbone with a much smaller model? For example, use llava-7b as the backbone of the draft model for the target model llava-13b. Will this further improve the efficiency?

---

> ### Author Response · Authors · 2025-11-20
> **Response to Reviewer Lox2**
>
> Thank you for the valuable comments. We add additional ablations to address these points and will integrate the results into the revision. Please let us know if further clarification or new analyses would be helpful.
>
> **More tasks:** Visual instruct tuning contains multiple tasks including fine-grained image captioning, ScienceQA, and commonsense reasoning. We further test the ability on each benchmark. Results show that FLASH consistantly overperforms the competing methods:
>  | Method\Task| COCO Captioning |  MMT-Bench| ScienceQA|
> |----------|----------|----------|----------|
> |EAGLE-MM| 2.28 ×  | 2.26×  |2.15 × |
> |DREAM| 2.49×   | 2.23× | 2.11× |
> |FLASH | 2.64  × | 2.38 × | 2.32× |
>
> **Novelty:** FLASH is more than a combination of existing techniques. While speculative decoding itself is a known mechanism, the core contributions of FLASH lie in how speculative decoding is adapted and made effective for vision–language models, which introduces several technical innovations that are not present in prior work. First, unlike previous multimodal speculative decoding methods that rely on fixed average pooling, FLASH introduces a learnable query-based visual compression module that selects the most informative visual features for speculative decoding. This design allows to aggressively reduce visual tokens while preserving semantics, leading to a substantially better accuracy–latency trade-off than all prior compression strategies (as shown in Table 4). Second, FLASH proposes a semi-autoregressive draft head that generates multiple draft tokens in parallel while remaining compatible with the target model’s verification behavior. Existing speculative decoding works for VLMs still rely on strictly autoregressive draft generation; our semi-autoregressive design is new, effective, and contributes significantly to the observed acceleration (as shown in Tables 1 and 2).
> Therefore, these components are not a straightforward reuse of existing techniques; rather, they are specifically tailored to the characteristics of vision–language tasks. Our ablation results further demonstrate that without these modality-specific designs, the method fails to achieve competitive speed-ups.
>
> **Draft Model Architecture:** The draft model is not a fine-tuned version of the target model. Instead, the draft head is a single-layer Transformer, which reuses the second-to-top layer feature of the target model as its input. It ensures strong compatibility between draft logits and target logits. We will clarify this architectural choice in the revision.
>
> **Draft Backbone:** We experimented with this idea early in development and found fundamental challenges. First, the target model’s intermediate hidden states (13B) do not match those of a smaller model (7B). Bridging them requires a learned projection or re-alignment module, which incurs additional parameters and latency. Second, speculative decoding relies on the draft model being close to the target model’s distribution. Draft–target compatibility is more important than draft model size. A different backbone inevitably leads to mismatched token preferences and inconsistent alignment.
> To directly address the reviewer’s concern about varying draft-model size, we presents the experimental results in our ablation on quantized draft model (c.f. Table 7 in Appendix).
> We also conducted ablation studies by varying the scale of the draft model, including the number of attention heads and the number of attention layers. The speed-up ratios under different scales are reported as follows:
> | Method\Task      | Visual Instruct Tuning | Video Captioning |
> |------------|-------------------|------------|
> |Head-32/Layer-1(FLASH)              |2.55×     | 1.83×    |
> |Head-64/Layer-1    | 2.56×     | 1.80×   |
> |Head-16/Layer-1|2.11×     | 1.34×     |
> |Head-32/Layer-2| 1.97×     | 1.55×     |
> These results suggest that larger draft models do not necessarily translate to better speed-performance trade-offs, especially in multimodal scenarios where visual token processing is already computationally demanding. Therefore, the speedup achieved by FLASH is not due to using a larger draft, but rather from design innovations like semi-autoregressive decoding and visual token compression. These analyses will be added in the revision.

---

> ### Comment · Area_Chair_uUcq · 2025-11-28
>
> Dear Reviewer, the discussion period is about to close. We kindly ask you to participate in the discussion or update your score based on the authors' rebuttal before the deadline. Thank you for your time and valuable contribution!

---

### Official Review · Reviewer_4Wgs · 2025-10-31

**Soundness:** 2
**Presentation:** 2
**Contribution:** 3
**Rating:** 4
**Confidence:** 2

**Summary:**

This paper introduces FLASH, a novel speculative decoding framework specifically designed for Large Multimodal Models (LMMs). The authors identify that existing speculative decoding methods, primarily developed for text-only LLMs, fail to leverage the unique properties of multimodal data. FLASH addresses this gap through two key innovations: (1) a visual token compression module that reduces redundant visual tokens from N to C tokens using learnable queries, and (2) a semi-autoregressive decoding head that generates K candidate tokens in parallel rather than sequentially. The method is evaluated on video captioning (Kinetics-400) and visual instruction tuning (LLaVA-instruct-150k) tasks using LLaVA-1.5 and QwenVL-2.5 as target models, achieving up to 2.68× speedup on video captioning and 2.55× on visual instruction tuning.

**Strengths:**

1. **Well-motivated approach**: The paper clearly identifies limitations of existing speculative decoding methods when applied to multimodal settings and provides compelling motivation for why multimodal-specific optimizations are needed.

2. **Novel technical contributions**: The combination of visual token compression and semi-autoregressive decoding is novel and well-suited to the characteristics of multimodal data. The observation about visual object co-occurrence enabling parallel token generation is insightful.

3. **Comprehensive experimental evaluation**: The experiments cover multiple models (LLaVA, QwenVL), multiple tasks (video captioning, visual instruction tuning), and multiple model sizes (3B to 32B parameters). The inclusion of both greedy (τ=0) and sampling (τ=1) decoding settings strengthens the evaluation.

4. **Strong empirical results**: FLASH consistently outperforms baseline methods, achieving substantial speedups (up to 2.68×) while maintaining output quality. The method shows particularly strong improvements over text-only speculative decoding approaches.

5. **Thorough ablation studies**: The paper includes useful ablations on different visual compression methods, the impact of K values, and performance in text-only scenarios, providing insights into each component's contribution.

**Weaknesses:**

1. **Limited theoretical analysis**: The paper lacks theoretical guarantees about the preservation of output distribution after semi-autoregressive decoding. While empirical results suggest quality is maintained, formal analysis would strengthen the claims.

2. **Insufficient comparison with concurrent work**: The paper primarily compares against adaptations of LLM speculative decoding methods (Eagle-MM, Medusa-MM) rather than other multimodal-specific acceleration techniques beyond Dream and MASSV.

3. **Unclear training details**: Critical implementation details are missing, such as:
   - How the compression ratio C/N is determined
   - Training time and computational requirements
   - Convergence behavior of the semi-autoregressive head

4. **Limited analysis of failure modes**: The paper doesn't discuss scenarios where FLASH might perform poorly or when the acceptance rate drops significantly. Understanding these limitations would be valuable for practitioners.

5. **Scalability concerns**: While experiments show results up to 32B parameters, the scalability to even larger models (70B+) or longer contexts is unclear. The fixed compression ratio might become problematic for very long videos.

**Questions:**

1. **How sensitive is FLASH to the compression ratio C/N?** The paper mentions reducing 576 tokens to 64 for LLaVA, but how was this ratio determined? What happens with different compression ratios?

2. **How does the semi-autoregressive head handle variable-length outputs?** When describing different visual regions, the phrase lengths might vary significantly. How does the fixed K affect performance in such cases?

3. **What is the impact on hallucination?** Visual token compression might lose important details. Have the authors evaluated whether FLASH increases hallucination rates compared to the original model?

4. **Can the method extend to other modalities?** The paper focuses on vision-language models. Could FLASH be adapted for audio-visual or other multimodal combinations?

5. **Why not apply compression to text tokens as well?** The paper only compresses visual tokens. Is there a fundamental reason why text token compression wouldn't work, or was this not explored?

6. **How does FLASH perform on fine-grained visual tasks?** Tasks requiring detailed visual understanding (e.g., OCR, diagram understanding) might be sensitive to token compression. Has this been evaluated?

7. **What is the memory overhead?** While inference time is reduced, what is the additional memory requirement for storing the draft model components?

---

> ### Author Response · Authors · 2025-11-20
> **Response to Reviewer 4Wgs**
>
> Thank you for the valuable comments. We add additional ablations to address these points and will integrate the results into the revision. Please let us know if further clarification or new analyses would be helpful.
>
> **Theoretical analysis:** Here we clarify that the draft model with semi-autoregressive decoding strictly follows the standard speculative decoding framework, whose theoretical properties have been established in prior work (Leviathan et al., 2023;Chen et al., 2023). In speculative decoding, although the draft model proposes multiple candidate tokens, the target model always performs a verification step. This verification step ensures that the final accepted tokens are sampled from exactly the same distribution as the target model’s original autoregressive distribution, regardless of how the draft model proposes them.
> FLASH modifies only the visual-token compression and the number of draft tokens per forward pass, without altering the acceptance rule. Therefore, the theoretical guarantee still holds that FLASH preserves the target model’s output distribution exactly.
>
> **Comparison with MASSV:** Thank you for pointing this out. We already compare with Dream and will additionally include results for MASSV.
>
> **Training details-compression ratio:** Regarding the compression ratio C/N, our main experiments use a ratio of 1/9, as reported in Table 4. It was selected empirically to balance speed-up and draft quality. To further address your concern, we additionally evaluate FLASH under C/N = 1/4 and C/N = 1/16, and report the speed-up ratios below:
> | C/N | k'  | Speed-up ratio |
> |----------|----------|----------|
> |1 (w/o compression) | 4 | 2.49 |
> |1/4 |4 | 2.51 |
> |1/9 (FLASH) | 4 |2.55 |
> |1/16 | 4 | 2.33 |
> |0 (w/o image) | 4 | 1.92 |
> These results show that FLASH remains effective across a broad range of compression ratios.
>
> **Training details-cost:** We will add the following discussion on the training time and memory overhead in revision: FLASH introduces only a lightweight draft head, which contains <0.5 GB of parameters. During inference, the dominant memory consumption comes from the target model’s KV cache and verification forward passes, while the draft head itself is negligible in comparison. The draft module is trained efficiently. In our experiments, training requires 10 hours on 2×NVIDIA V100 GPUs.
>
> **Training details-convergence:** Under speculative decoding, the convergence of the semi-autoregressive draft head is reflected directly through the average acceptance tokens and the resulting overall speed-up ratios. FLASH consistently achieves a higher speed-up ratio compared with competing methods. This indicates that the semi-autoregressive head has indeed converged to a stable approximation of the target model’s token distribution. In other words, the improved speed-up is an indirect and strong signal that the draft head is well-aligned and well-trained.
>
> **Failure cases:** Semi-autoregressive head may brings local prediction errors, such as repeated words (which is expected given that it does not utilize full temporal information during prediction). However, these failure modes only affect the speed-up ratio, not the correctness, because speculative decoding ensures that every output token is validated against the target model. More importantly, as shown in Table 1 and 2, the overall speed-up ratio remains superior. This indicates that although semi-autoregressive drafting and visual compression may reduce draft accuracy, the gain in efficiency more than compensates for such local losses. Discussion will be added.

---

> ### Author Response · Authors · 2025-11-20
> **Response to Reviewer 4Wgs (continuation)**
>
> **Scalability:** Regarding scalability, prior speculative decoding works for multimodal LLMs (e.g., DREAM) only report results up to 13B. In this paper, we further extend the analysis to 32B, already substantially larger than previous studies. Empirically, the trends observed across 7B to 13B and to 32B indicate that the speed-up ratio of FLASH scale smoothly with model size, and we do not observe new bottlenecks emerging at larger scales. This suggests that scaling to even larger models (e.g., 70B+) is unlikely to introduce qualitatively different behavior. In addition, for fair comparison, prior work conducts benchmarking using a single GPU, and we follow this evaluation protocol to ensure comparability. This setup does not practically allow testing 70B-level models under the same constraints. Nevertheless, the speculative decoding mechanism itself does not impose architectural limits on scaling to larger models; the main constraint is hardware rather than methodology.
>
> **Long context and videos:** The speculative decoding itself does not impose fundamental limits on sequence length. The draft model only generates the next k draft candidates at each step, so handling extremely long contexts is primarily a concern for the target model, not the draft. Consequently, handling extremely long sequences is primarily determined by the target model’s KV-cache capacity, rather than by the speculative-decoding mechanism or FLASH’s design.
>
> **Hallucination:** As we pointed in Theoretical analysis above, all draft tokens must pass a strict token-by-token verification using the target model logits, identical to standard speculative decoding. In other words, FLASH does not alter, approximate, or relax the target distribution; it only proposes candidates more efficiently. Thus, the final output distribution of FLASH is theoretically preserved and equivalent to that of the target model. In practice, if the draft model generates hallucinated tokens, the target model will reject them during verification, which will reduce the overall speed-up ratio. FLASH achieves higher speed-up compared to prior methods, indicating that fewer draft tokens are rejected on average. From this perspective, FLASH effectively reduces the occurrence of hallucinated proposals while maintaining the same output fidelity as the target model.
>
> **Other modalities:** Although speculative decoding is theoretically modality-agnostic and can be applied to any sequence of embeddings, practical deployment requires modality-specific architectural design. This is similar to how NLP techniques do not directly transfer to vision–language models without substantial modifications (e.g., our proposed semi-autoregressive head and visual token compression). In our case, extending FLASH from vision–language to audio (or other modalities) would face analogous challenges. Audio exhibits very different temporal structure, redundancy patterns, and alignment behaviors compared to visual tokens. As a result, the key components that enable FLASH’s speedup—such as our learnable visual-token compression and the semi-autoregressive draft head—might not transfer directly. New, audio-specific designs would be required to achieve meaningful acceleration. Therefore, FLASH’s practical instantiation is tailored to VLMs, and extending it to audio is non-trivial and outside the scope of this work.
>
> **Compression on text tokens:** Compared with visual embeddings, text tokens are considerably harder to compress: they are discrete, highly information-dense units produced by a tokenizer, and even small perturbations can noticeably affect the resulting language distribution.
> To support this claim, we additionally experimented with compressing text tokens using the same compression module used for vision:
> | Method | C/N on text  | Speed-up Ratio |
> |----------|----------|----------|
> |Average Pooling |1/4| 0.83|
> |Average Pooling |1/2| 0.88 |
> |FLASH | 1/4 | 0.89 |
> |FLASH | 1  | 2.55 |
> The results show severe degradation on speed-up ratios, confirming that text compression is not feasible under current architectures. By contrast, multimodal models contain hundreds to thousands of visual tokens, many of which are redundant. This high redundancy is unique to VLMs and is precisely why visual-token compression substantially improves speed without harming quality, while text compression does not.
>
> **More tasks:** Visual instruct tuning contains multiple tasks including fine-grained image captioning, ScienceQA, and commonsense reasoning. We further test the ability on each benchmark. Results show that FLASH consistantly overperforms the competing methods:
>  | Method\Task | COCO Captioning |  MMT-Bench| ScienceQA|
> |----------|----------|----------|----------|
> |EAGLE-MM| 2.28 ×  | 2.26×  |2.15 × |
> |DREAM| 2.49×   | 2.23× | 2.11× |
> |FLASH | 2.64  × | 2.38 × | 2.32× |

---

> ### Comment · Area_Chair_uUcq · 2025-11-28
>
> Dear Reviewer, the discussion period is about to close. We kindly ask you to participate in the discussion or update your score based on the authors' rebuttal before the deadline. Thank you for your time and valuable contribution!

---

### Official Review · Reviewer_iSr7 · 2025-11-02

**Soundness:** 3
**Presentation:** 3
**Contribution:** 3
**Rating:** 6
**Confidence:** 4

**Summary:**

The manuscripts proposes a novel framework for accelerating the decoding process of large multi-modal models, which uses a draft model to produce multiple next candidate tokens before they are verified by the target model. Prior works for efficient decoding in LMM paradigm often choose text-only draft models, neglecting that visual inputs contain critical information for the subsequent decoding process. To address this, the manuscript presents FLASH, generating candidate tokens using compressed visual tokens and a semi-autoregressive decoding strategy. The visual tokens are compressed due to the high redundancy in visual tokens, and multiple tokens are generated at once using the semi-autoregressive decoding strategy, instead of generating tokens one by one, to further speed up the decoding process. Overall, FLASH achieves a strong performance compared to previous speculative decoding methods.

**Strengths:**

1. The proposed method, FLASH, not only takes into account the importance of visual information in speculative decoding, but also reduces the redundancy in the visual representations for improved efficiency.

2. Multi-token prediction strategy is employed for further speeding up the draft generation process. Though both of the ideas are hardly novel, the combined usage in the speculative decoding paradigm is crucial for the field.

3. Quantitative evaluation of FLASH reveals a high acceptance rate, hence an improved speed-up ratio.

**Weaknesses:**

1. Though FLASH seems effective when compared to existing approaches, the component-wise analysis is not sufficient. For example, I would be curious to know, compared to no speculative decoding strategy, how does FLASH perform under different compression strategies, different number of visual tokens after compression, different number of tokens produced per-forward-pass by the draft model, etc.

2. Since the draft model in FLASH is capable of generating multiple tokens at one single forward pass, is the metric 'average acceptance tokens' still effective for comparing the effectiveness or the performance of different decoding strategies?

3. It is not evaluated or explained how the decoding strategy affects final benchmark performance.

**Questions:**

1. Just for discussion, how does the size of the draft model affect the acceptance rate and the system-level speed-up rate of LMMs? (From my understanding, if the model produces draft tokens slowly but more accurately, the system-level speed-up rate would be higher, is it correct?)

---

> ### Author Response · Authors · 2025-11-20
> **Response to  Reviewer iSr7**
>
> Thank you for the insightful comments. We have conducted additional ablations to address these concerns and will incorporate them in the revision.
>
> **w/o speculative decoding:**
> To clarify the contribution of speculative decoding itself, we add a baseline that removes the verification mechanism entirely: the draft model directly outputs the full sequence without any target-model verification (i.e., purely relying on the draft). The results are shown as follows:
> | Method | Accuracy | Draft length |
> |--------|----------|--------------|
> | w/o SD | 0.069    | 192          |
> | FLASH  | 0.648    | 4            |
> Accuracy measures whether the text generated by the draft model is consistent with it generated by the target model. This result confirms that accuracy collapses due to the cumulative error without verification. It demonstrates that the speculative decoding mechanism is necessary for correctness and cannot be relied simply on a lightweight draft model.
>
> **Ablations on compression strategies and draft length:** Please refer to Table 4 for the ablations on different visual compression strategies. Following your advices, we further conduct ablations covering the number of visual tokens after compression, and the number of draft tokens produced per forward pass of the draft model k'. The results are shown as follows:
> | Compression ratios C/N | k'  | Speed-up ratio |
> |----------|----------|----------|
> |1 (w/o compression) | 4 | 2.49 |
> |1/4 |4 | 2.51 |
> |1/9 (FLASH) | 4 |2.55 |
> |1/16 | 4 | 2.33 |
> |0 (w/o image) | 4 | 1.92 |
> |1/9  | 2 | 2.47 |
> |1/9 | 1 (autoregressive) | 2.23 |
> These results show that the chosen compression ratio (1/9) offers a balance of efficiency and draft quality. Besides, the speed-up ratio remains generally robust across different compression ratios (C/N), with only the extreme case of removing the image entirely causing a clear degradation.
>
> **Average acceptance tokens:** In multi-token-per-step speculative decoding, average accepted tokens remains a direct and interpretable indicator of semi-autoregressive efficiency. It quantifies the efficiency of the draft model, by evaluating how many tokens per step are adopted. We will clarify this definition and explicitly connect it to the overall speed-up ratios.
>
> **Decoding strategy:** We additionally evaluate how different decoding hyperparameters influence the overall speed-up ratios. In particular, we varied the semi-autoregressive draft length k′ (k′ is the number of tokens generated per forward pass by the draft model). Our main results were reported with k' = K (K is the number of candidate tokens for each verification).
> Across different values of k', we observe that the speed-up ratio is stable (results are listed in the Table above). This is because speculative decoding primarily relies on the consistency between the draft distribution and the target model, while the specific decoding strategy (e.g., choosing different k' or autoregressive pattern) does not alter the target model’s verification behavior. The target model only accepts tokens that match its distribution, so the decoding strategy only affects the overall speed-ups while the output remains consistent with the target model.
> To ensure we fully address your concern, we would appreciate clarification on which specific "decoding strategy" you were referring to, and we are happy to provide additional experiments and analyses.
>
> **Draft model size:** Your intuition is correct that a larger draft model generally produces more accurate draft tokens, thus having a higher acceptance rate. However, the overall acceleration exhibits a fundamental trade-off that a larger draft model has slower draft inference, thus having low speed-up ratios. A theoretical extreme illustrates this clearly: if the draft model were as large as the target model, the acceptance rate would approach 100%, but the total computational cost would be draft forward + target verification > standard decoding, which exceeds the cost of standard autoregressive decoding. Thus the speed-up becomes < 1.
> In FLASH, we explore this trade-off via a quantized draft model (c.f. Table 7). And we also test the results on different scales of the draft model, including the number of attention heads and the number of attention layers:
> | Method\Task      | Visual Instruct Tuning | Video Captioning |
> |--------------------|-----------|------------|
> |Head-32/Layer-1(FLASH)              |2.55×     | 1.83×    |
> |Head-64/Layer-1    | 2.56×     | 1.80×   |
> |Head-16/Layer-1|2.11×     | 1.34×     |
> |Head-32/Layer-2| 1.97×     | 1.55×     |
> These results show that balance between the scale of draft model and speed is important. For example, a draft model with 2 Transformer layers may have more accuracy in predicting the target distributions. However, at the same time, it introduces additional computational cost thus affect the overall speed-up ratios.
> We will include these ablations in the revised manuscript.

---

> ### Comment · Area_Chair_uUcq · 2025-11-28
>
> Dear Reviewer, the discussion period is about to close. We kindly ask you to participate in the discussion or update your score based on the authors' rebuttal before the deadline. Thank you for your time and valuable contribution!

---

### Author Response · Authors · 2025-11-28
**Comment to Reviewers**

Thank you for your thoughtful review. We hope our rebuttal has addressed the concerns you raised. If anything remains unclear or if you have additional questions, we would be happy to clarify further. Please let us know whether our responses resolve your earlier concerns.

---

### Author Response · Authors · 2025-12-02
**Author Summary of Rebuttal**

We thank the reviewers for their detailed feedback. We appreciate their recognition of our novel combination of latent-aware visual token compression and semi-autoregressive speculative decoding, which together accelerate multimodal LLM inference.

**Strengths recognized by reviewers.**

Reviewers highlighted the practical impact and efficiency gains, with clear experimental evidence showing significant speed-up while maintaining output quality. The method was noted as well-motivated, clearly explained, and reproducible, demonstrating thoughtful design and strong engineering rationale.

Below we summarize how our rebuttal has addressed the main concerns and clarified several misunderstandings.

**Addressed reviewer concerns.**
1. More tasks of evaluation: We report the overall speed-up ratios on visual instruct tuning (which covers multiple tasks) in the Main Text, and we additionally provide results for several individual tasks in Appendix, such as ScienceQA.

2. Scale of draft model: Results for the quantized draft model are reported in the Main Text, and we additionally provide results obtained by varying the draft model’s parameters, including the number of layers and attention heads.

3. Ablations on compression ratios: We provide additional results on different compression ratios in revision. And the results show that the speed-up ratio remains generally robust across different compression ratios.


**Clarified misunderstandings.**
1. Quality of the draft candidates: We emphasize that the verification step in speculative decoding ensures the final output remains aligned with the target model. The quality of the draft model directly impacts the overall speed-up ratios: low-quality draft candidates (including hallucinations or misalignments with the target model) will be rejected, resulting in lower speed-up ratios.

2. Motivation of model design: We clarify the motivation and design of both visual-token compression and semi-autoregressive draft generation, and provide explanations demonstrating why they are effective in multimodal tasks.


Given the novelty of combining latent-aware compression with semi-autoregressive speculative decoding for multimodal tasks, and considering that all reviewer concerns were either addressed or clarified, we hope the AC will find our responses satisfactory and consider the manuscript favorably.

---

### Meta-Review · Area_Chair_magw · 2026-01-06

**Summary:**

The reviewers raise some concerns about the limited theoretical analysis (4Wgs), limited novelty (Lox2,ht7i), unclear motivation (ht7i), insufficient comparison and evaluation (4Wgs,Lox2,iSr7), missing training details (4Wgs), scalability concerns (4Wgs), and inconsistencies (ht7i). Before rebuttal, three reviewers graded the paper marginally below acceptance, while one marginally above.

**Reviewer Concerns:**

The authors provided missing details and explanations, including details related to speedups. Related to the novelty, the work still feels like being a combination of existing approaches, and while the approach builds on top of established theory, the link between them from a theoretical perspective, which could have been the element of novelty, is not sufficiently motivated.

**Reviewer Scores:**

Although the authors put consistent effort in their rebuttal and clarifies some of the shady points, there is still a gap in the justification and and novelty of the approach that should be worked on. The feeling is that the final evaluation would have overall stayed borderline leaning to reject.

---

### Decision · Program_Chairs · 2026-01-26

Reject